

# *Elatostema qinzhouense* (Urticaceae), a new species from limestone karst in Guangxi, China

Longfei Fu[1,2], Alexandre K. Monro[3], Tiange Yang[4], Fang Wen[2], Bo Pan[2], Zibing Xin[2], Zhixiang Zhang[1] and Yigang Wei[2]

[1] Laboratory of Systematic Evolution and Biogeography of Woody Plants, College of Nature Conservation, Beijing Forestry University, Beijing, China
[2] Guangxi Key Laboratory of Plant Conservation and Restoration Ecology in Karst Terrain, Guangxi Institute of Botany, Guangxi Zhuang Autonomous Region and Chinese Academy of Sciences, Guilin, China
[3] Identification & Naming Department, Royal Botanic Gardens, Kew, London, UK
[4] College of Life Sciences & Key Laboratory for Protection and Application of Special Plant Germplasm in Wuling Area of Hubei Province, South-Central University for Nationalities, Wuhan, China

Corresponding authors
Zhixiang Zhang, zxzhang@bjfu.edu.cn
Yigang Wei, weiyigang@aliyun.com

## ABSTRACT

*Elatostema qinzhouense* L.F. Fu, A.K. Monro & Y.G. Wei, a new species from Guangxi, China is described and illustrated. Morphologically, *E. qinzhouense* is most similar to *E. hezhouense* from which it differs by having smaller size of leaf laminae, fewer and smaller staminate peduncle bracts, longer pistillate peduncle bracts and a larger achene. This result is supported by the molecular evidence. The phylogenetic position of the new species within *Elatostema* is evaluated using three DNA regions, ITS, *trnH-psbA* and *psbM-trnD*, for 107 taxa of *Elatostema* s.l. (including *E. qinzhouense*). Bayesian inference (BI) and maximum likelihood (ML) analyses each recovered the same strongly supported tree topologies, indicating that *E. qinzhouense* is a member of the core *Elatostema* clade and sister to *E. hezhouense*. Along with the phylogenetic studies, plastid genome and ribosomal DNA (rDNA) sequences of the new species are assembled and annotated. The plastid genome is 150,398 bp in length and comprises two inverted repeats (IRs) of 24,688 bp separated by a large single-copy of 83,919 bp and a small single-copy of 17,103 bp. A total of 113 functional genes are recovered, comprising 79 protein-coding genes, 30 tRNA genes, and four rRNA genes. The rDNA is 5,804 bp in length and comprised the 18S ribosomal RNA partial sequence (1,809 bp), internal transcribed spacer 1 (213 bp), 5.8S ribosomal RNA (164 bp), internal transcribed spacer 2 (248 bp) and 26S ribosomal RNA partial sequence (3,370 bp). In addition, the chromosome number of *E. qinzhouense* is observed to be $2n = 26$, suggesting that the species is diploid. Given a consistent relationship between ploidy level and reproductive system in *Elatostema*, the new species is also considered to be sexually reproducing. Our assessment of the extinction threat for *E. qinzhouense* is that it is Endangered (EN) according to the criteria of the International Union for Conservation of Nature.

## INTRODUCTION

*Elatostema* J.R.Forst. & G.Forst. is one of two species-rich genera in the Urticaceae, the other being *Pilea* Lindl. *Elatostema* comprises several hundred species of succulent herbs and subshrubs that grow in shade in forests, stream sides, gorges and caves (*Fu et al., 2017a*; *Monro et al., 2018*). *Elatostema* is distributed throughout tropical and subtropical Africa, Madagascar, Asia, Australia and Oceania (*Lin, Friis & Wilmot-Dear, 2003*).

The delimitation of *Elatostema* has long been controversial with respect to *Elatostematoides* C.B.Rob., *Pellionia* Gaudich. and *Procris* Comm. ex Juss. A recent analysis of molecular and morphological evidence suggests that *Elatostema* is monophyletic and includes taxa attributed to *Pellionia* but excludes those attributed to *Elatostematoides*, *Procris*, and *Pellionia repens* (Lour.) Merr. (*Tseng et al., 2019*). The latest revision of Chinese *Elatostema* (*Wang, 2014*) indicates China, with more than 280 species, as the center of diversity for the genus but this may be an artefact of sampling and taxonomic effort (*Fu et al., 2019a*). Circa 2/3 (184 species) of Chinese *Elatostema* are associated with the limestone karst of Guangxi, Guizhou and Yunnan in Southwest China (*Wang, 2014*) and it is likely that limestone karst is an important source of species diversity and point endemics throughout its range.

*Fu et al. (2019b)* demonstrate that a large proportion of *Elatostema* species are known from a single collection, which increases the risk of the over-description of species (*Wei, Monro & Wang, 2011*). DNA sequence data provides molecular evidence to confirm the relationship of undescribed species to described taxa and so can help mitigate this risk by enabling the closely related species to be identified (*Fu et al., 2019b*; *Wells et al., 2021*). For Chinese species, sequence data for 60 species (21% of the flora) are available for this purpose (LF Fu, 2021, pers. obs.).

Cytological data can also provide additional evidence to confirm generic placement in Urticaceae (*Subramanian & Thilagavathy, 1988*; *Kanemoto & Yokota, 1998*; *Yamashiro et al., 2000*; *Kanemoto et al., 2015*). Within *Elatostema*, it has also been successfully used for recognizing hybrid taxa (*Tseng & Hu, 2014*), as well as inferring ploidy and reproductive systems (*Fu et al., 2017b*).

In 2014, while conducting field work in Guangxi, China, we found a hitherto undescribed species of *Elatostema*. The population comprised sterile plants with a distinctive and unfamiliar morphology. In order to obtain fertile plants of this taxon for identification, several individuals were collected to be introduced at Guilin Botanical Garden for further study. Once flowering, we were able to undertake a thorough literature survey and review of herbarium specimens at BM, IBK, IBSC, K and PE, along with the molecular and cytological studies. These confirmed that the material was of a hitherto undescribed species.

## MATERIALS & METHODS

### Ethics statement

All the collecting locations of the new species reported in this study are outside any natural conservation area and no specific permissions were required for these locations. Since the species are currently undescribed, they are not currently included in the China Species Red

List (*Wang & Xie, 2004*). Our field studies did not involve any endangered or protected species. No specific permits were required for the present study.

## Nomenclature

The electronic version of this article in Portable Document Format (PDF) will represent a published work according to the International Code of Nomenclature for algae, fungi, and plants (ICN), hence the new names contained in the electronic version are effectively published under that Code from the electronic edition alone. In addition, new names contained in this work which have been issued with identifiers by IPNI will eventually be made available to the Global Names Index. The IPNI can be accessed and the associated information contained in this publication viewed through any standard web browser using the web address http://ipni.org/. The online version of this work is archived and available from the following digital repositories: PeerJ, PubMed Central, and CLOCKSS.

## Morphological examination

We used the morphological species concepts of *Wei, Monro & Wang (2011)*, *Fu et al. (2014)*, *Wang (2014)* and *Fu et al. (2017a)* to distinguish and compare taxa, placing emphasis on peduncle bract shape and length, the number, morphology and arrangement of the bracts comprising the receptacle-like involucre, the number of bracteoles per flower and leaf lamina length/width ratios. Material was examined using an Olympus SZX16 binocular microscope (Japan) and Plan Apo lens at × 10 and × 90 magnifications (*Wei, Monro & Wang, 2011*). For achene morphology, we also made scanning electron micrograph (SEM) observations. Fruiting material was collected from specimens, dried, and mounted using double-sided adhesive tape and coated with gold using a sputter coater. The fruit were then observed and photographed under a ZEISS EVO18 scanning electron microscope. At least five achenes were used to determine their size and surface ornamentation.

## Extinction threat assessment

An extinction threat assessment was undertaken for the new species described here using IUCN criteria (*IUCN, 2001*; *IUCN, 2019*). Calculations of the extent of occurrence (EOO) and area of occupation (AOO) were undertaken using the online conservation assessment tool GeoCATAT (*Bachman et al., 2011*). The AOO was calculated using a cell width of 2 km as recommended by *IUCN (2019)*.

## Genomic DNA extraction and sequencing

Leaf material for DNA extraction was dried using silica gel (*Chase & Hills, 1991*). Genomic DNA was extracted using a modified CTAB protocol (*Chen et al., 2014*) and assessed by agarose gel electrophoresis. The total gDNA sample was sent to Majorbio (http://www.majorbio.com/, China) for library construction and next-generation sequencing. Short-insert (350 bp) paired-end read libraries preparation and 2 × 150 bp sequencing were performed on an Illumina (HiSeq4000) genome analyzer platform. Approximately 2 Gb of raw data for the new species was filtered using the FASTX-Toolkit to obtain high-quality clean data by removing adaptors and low-quality reads (http://hannonlab.cshl.edu/fastx_toolkit/download.html).

## Plastid genome and ribosomal DNA (rDNA) assembly and annotation

Clean reads were paired and imported in Geneious Prime (*Kearse et al., 2012*). For plastid genome assembly, the clean reads were mapped to published plastid genome sequence as reference (*Fu et al., 2019c*) using the Fine Tuning option in Geneious Prime (iterating set as 10 times) to exclude nuclear and mitochondrial reads. Then, *de novo* assembly was performed using Geneious Prime with a medium-low sensitivity setting to assemble plastid genome sequence. The generated contigs was mapped by the clean reads using the Fine Tuning option in Geneious Prime (iterating set as 10 times) to fill gaps. Contigs were able to be concatenated using the Repeat Finder option implemented in Geneious Prime until a ~130 kb contig (including SSC, IR and LSC) being built. The Inverted repeat (IR) region was determined by the Repeat Finder option in Geneious Prime and was reverse copied to obtain the complete plastid genome. The annotation approach of plastid genome was performed using CPGAVAS2 and PGA (*Qu et al., 2019*; *Shi et al., 2019*). The process of rDNA assembly is generally same to plastid genome assembly with the exception of different reference (*Gryta et al., 2017*) and iterating as none. The annotation approach of rDNA was performed using Annotate option in Geneious Prime.

## Phylogenetic analyses

We generated a phylogeny using sequences data from previous phylogenies of *Elatostema* s.l. (*Tseng et al., 2019*). We extracted three DNA regions (ITS, *trnH-psbA* and *psbM-trnD*) from assembled rDNA and complete plastid genome sequences, respectively of the new species and downloaded most of the sequences data used in *Tseng et al. (2019)* from Genbank (details see Table S1). This resulted in 107 taxa (116 accessions) of *Elatostema* s.l. as ingroup and three species, belonging to *Lecanthus*, *Poikilopermum* and *Debregeasia*, as outgroup. Three datasets (ITS, *trnH-psbA* and *psbM-trnD*) were aligned independently using multiple alignment using fast Fourier transform (MAFFT) version 7.0 (*Katoh & Standley, 2013*) with default settings, followed by manual adjustment. The two best supported tree topologies from maximum likelihood (ML) analyses of cpDNA and nrITS were visually compared for topological incongruence. A conflict in tree topologies of each tree was considered significant when incongruent topologies both received bootstrap values ≥ 80% (*Monro, 2006*; *Tseng et al., 2019*). As there was no significant incongruence between two datasets (*Tseng et al., 2019*), phylogenies were reconstructed based on the combined dataset using ML and Bayesian inference (BI). The BI and ML analyses were performed followed *Tseng et al. (2019)*.

## Cytological experiments

Stem cuttings of the new species were grown in tap-water in a culture room. Actively growing root tips were harvested after 3–4 weeks. Cytological examination followed *Fu et al. (2017b)*. The best metaphase plates were photographed using an imager microscope with a camera attachment. At least 3–5 cells from 3–5 root tips of the new species at somatic metaphase were counted to determine the chromosome numbers.
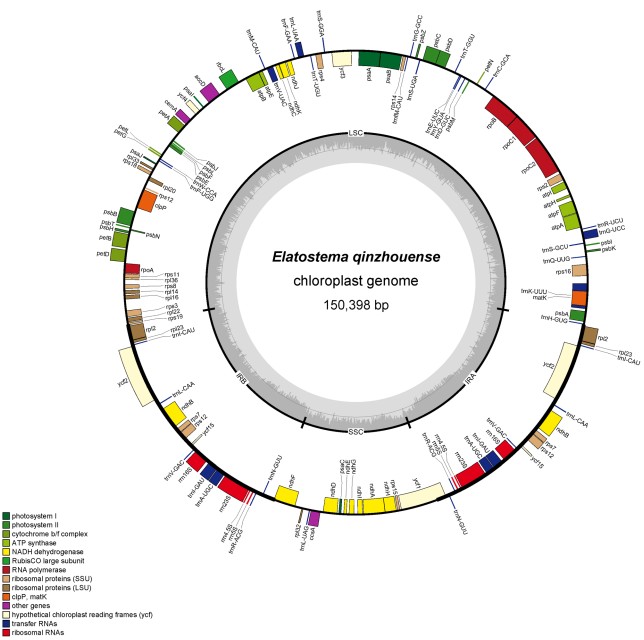

**Figure 1** **Plastid genome map of *Elatostema qinzhouense*.** The thick lines on the outer complete circle identify the inverted repeat regions (IRa and IRb). The innermost track of the plastome shows the GC content. Genes on the outside and inside of the map are transcribed in clockwise and counter directions, respectively.

## RESULTS

### Characteristics of the complete plastid genome and ribosomal DNA

The complete plastid genome and ribosomal DNA (rDNA) sequences of *Elatostema qinzhouense* comprised 150,398 bp (Fig. 1) and 5,804 bp, respectively. The characteristics and statistics of plastid genome and rDNA were summarized in Tables 1 and 2.

### Phylogenetic reconstruction

The characteristics and statistics of the datasets used in this study are presented in Table 3. BI and ML analyses of dataset of three DNA regions (ITS, *trnH-psbA* and *psbM-trnD*) resulted in the same tree topologies that both indicate the new species recovering in the clade of *Elatostema* (PP = 1, BP = 100%) and sister to *E. hezhouense* W.T. Wang, Y.G. Wei & A.K. Monro (PP = 1, BP = 100%) (Fig. 2).

### Chromosome characteristics

The chromosome number of *Elatostema qinzhouense* was observed to be 2n = 26. The chromosome size fell into the range 1.45–3.33 μm, suggesting slight size variation. Detailed karyotype analysis was not possible because the chromosomes were small and had unclear centromeres (Fig. 3).

### Taxonomic treatment

*Elatostema qinzhouense* **L.F. Fu, A.K. Monro & Y.G. Wei, sp. nov.** (Figs. 4 and 5 and S1)

**Table 1 Summary of plastid genome and rDNA of *Elatostema qinzhouense*.**

| | Characteristic | *Elatostema qinzhouense* |
|---|---|---|
| Plastid genome | Size (base pair, bp) | 150,398 |
| | LSC length (bp) | 83,919 |
| | SSC length (bp) | 17,103 |
| | IR length (bp) | 24,688 |
| | Number of genes | 113 |
| | Protein-coding genes | 79 |
| | rRNA genes | 4 |
| | tRNA genes | 30 |
| | LSC GC% | 33.75% |
| | SSC GC% | 29.82% |
| | IR GC% | 43.00% |
| rDNA | Size (bp) | 5,804 |
| | 18S ribosomal RNA partial sequence (bp) | 1,809 |
| | internal transcribed spacer 1 (bp) | 213 |
| | 5.8S ribosomal RNA (bp) | 164 |
| | internal transcribed spacer 2 (bp) | 248 |
| | 26S ribosomal RNA partial sequence (bp) | 3,370 |

## IPNI

**Type.** China. Guangxi: cultivated material in Guilin Botanical Garden harvested on 24 March 2017, wild-collected, from Taiping Town, Lingshan County, Qinzhou City, 22.408N, 108.835E (WGS84), elev. 124 m, 26 May 2014, *Pan B and Ma HS P1184* (holotype IBK (IBK00426150!); isotype K (K000798321!)).

**Diagnosis.** Most similar to *Elatostema hezhouense* from which it differs by the smaller size of leaf laminae (10–45 × 6–15 mm vs. 55–115 × 20–25 mm), fewer and smaller staminate peduncle bracts (1, 1 mm vs. 2, 3.5 mm), longer pistillate peduncle bract (0.900 mm vs. 0.375 mm) and a larger achene (0.86–0.94 × 0.27–0.30 mm vs. 0.6 × 0.25 mm) (see Table 4).

**Description.** Perennial herb, epipetric, monoecious. Not tuber forming. Stem 150–270 × 1–2 mm, erect to arching, branched, drying finely sulcate, furfuraceous, glabrous, with internodes 6–27 mm. Stipules solitary, opposite the leaf at each node, persistent, 2–3 × 0.75–1.5 mm, narrowly lanceolate, glabrous. Leaves distichous, alternate, petioles 1–1.5 × 0.5 mm, glabrous; laminae 10–45 × 6–15 mm, length:width ratio 1.6–3.5:1, asymmetrically oblanceolate or falcate, chartaceous, 5-nerved, the secondary nerves 4–6 pairs, 30–45° to the midrib; upper surface drying green, furfuraceous, glabrous, cystoliths densely scattered, fusiform, 0.2–0.5 mm; lower surface drying yellow-green, furfuraceous glabrous, sparsely brown glandular, cystoliths absent; base asymmetrical, broader-half auriculate, narrower-half cuneate; margin serrate-dentate, teeth 3–5 mm apart; apex acuminate. Staminate and pistillate inflorescences borne on the same stems concurrently. Staminate inflorescences solitary, 5–7 mm, bearing 2–4 flowers in a receptacle-like involucre; peduncle ca 0.9 × 0.5 mm, glabrous, bracteate; bract 1 mm, deltate; receptacle ca 2.5 × 1 mm, oblong, not
**Table 2  Genes encoded in plastid genome of *Elatostema qinzhouense*.**

| Group of genes | Gene name |
| --- | --- |
| tRNA genes | *trnH-GUG, trnK-UUU\*, trnQ-UUG, trnS-GCU, trnG-UCC\*, trnR-UCU, trnC-GCA, trnD-GUC, trnY-GUA, trnE-UUC, trnT-GGU, trnS-UGA, trnG-GCC, trnfM-CAU, trnS-GGA, trnT-UGU, trnL-UAA\*, trnF-GAA, trnV-UAC\*, trnM-CAU, trnW-CCA, trnP-UGG, trnI-CAU (×2), trnL-CAA (×2), trnV-GAC (×2), trnI-GAU\* (×2), trnA-UGC\* (×2), trnR-ACG (×2), trnN-GUU (×2), trnL-UAG* |
| rRNA genes | *rrn16* (×2), *rrn23* (×2), *rrn4.5* (×2), *rrn5* (×2) |
| Ribosomal small subunit | *rps16\*, rps2, rps14, rps4, rps18, rps12\*\** (×2), *rps11, rps8, rps3, rps19, rps7* (×2), *rps15* |
| Ribosomal large subunit | *rpl33, rpl32, rpl20, rpl36, rpl14, rpl16\*, rpl22, rpl2\** (×2), *rpl23* (×2) |
| DNA-dependent RNA polymerase | *rpoC2, rpoC1\*, rpoB, rpoA* |
| Photosystem I | *psaB, psaA, psaI, psaJ, psaC* |
| Large subunit of rubisco | *rbcL* |
| Photosystem II | *psbA, psbK, psbI, psbM, psbC, psbZ, psbG, psbJ, psbL, psbF, psbE, psbB, psbT, psbN, psbH* |
| NADH dehydrogenase | *ndhJ, ndhK, ndhC, ndhB\** (×2), *ndhF, ndhD, ndhE, ndhG, ndhI, ndhA\*, ndhH* |
| Cytochrome b/f complex | *petN, petA, petL, petG, petB\*, petD\** |
| ATP synthase | *atpA, atpF\*, atpH, atpI, atpE, atpB* |
| Maturase | *matK* |
| Subunit of acetyl-CoA carboxylase | *accD* |
| Envelope membrane protein | *cemA* |
| Protease | *clpP\*\** |
| C-type cytochrome synthesis | *ccsA* |
| Conserved open reading frames (ycf) | *ycf3\*\*, ycf4, ycf2* (×2), *ycf1, ycf15* (×2) |

**Notes.**
Genes with one or two introns are indicated by one (*) or two asterisks (**), respectively. Genes in the IR regions are followed by the (×2) symbol.

**Table 3  Statistics for the molecular datasets used in this study.**

| | Number of sequences (ingroup/ outgroup) | Aligned length (bp) | Length variation (bp) | Variable characters (bp) | Parsimony -informative characters (bp) | Model selected (AIC) |
| --- | --- | --- | --- | --- | --- | --- |
| ITS | 118/3 | 887 | 571–710 | 569 | 484 | – |
| *trnH-psbA* | 118/3 | 370 | 174–263 | 196 | 128 | – |
| *psbM-trnD* | 118/3 | 645 | 329–522 | 254 | 129 | – |
| Combined plastid | 118/3 | 1,015 | 503–767 | 450 | 257 | – |
| Combined all | 118/3 | 1,902 | 1,170–1,416 | 1,020 | 741 | GTR + I + G |

lobed, glabrous, subtended by marginal bracts, marginal bracts 6, unequal, outer major bracts 2, inner minor bracts 4; major bracts 4–5 mm, broadly ovate, the apex corniculate, minor bracts ca 5 mm, ovate. Staminate flowers 2.5–3 × 2 mm, white, pedicellate, pedicel 1.5–2.0 mm; bracteoles 2, equal, 2–3 mm, narrowly lanceolate or linear; tepals 4 or 5,

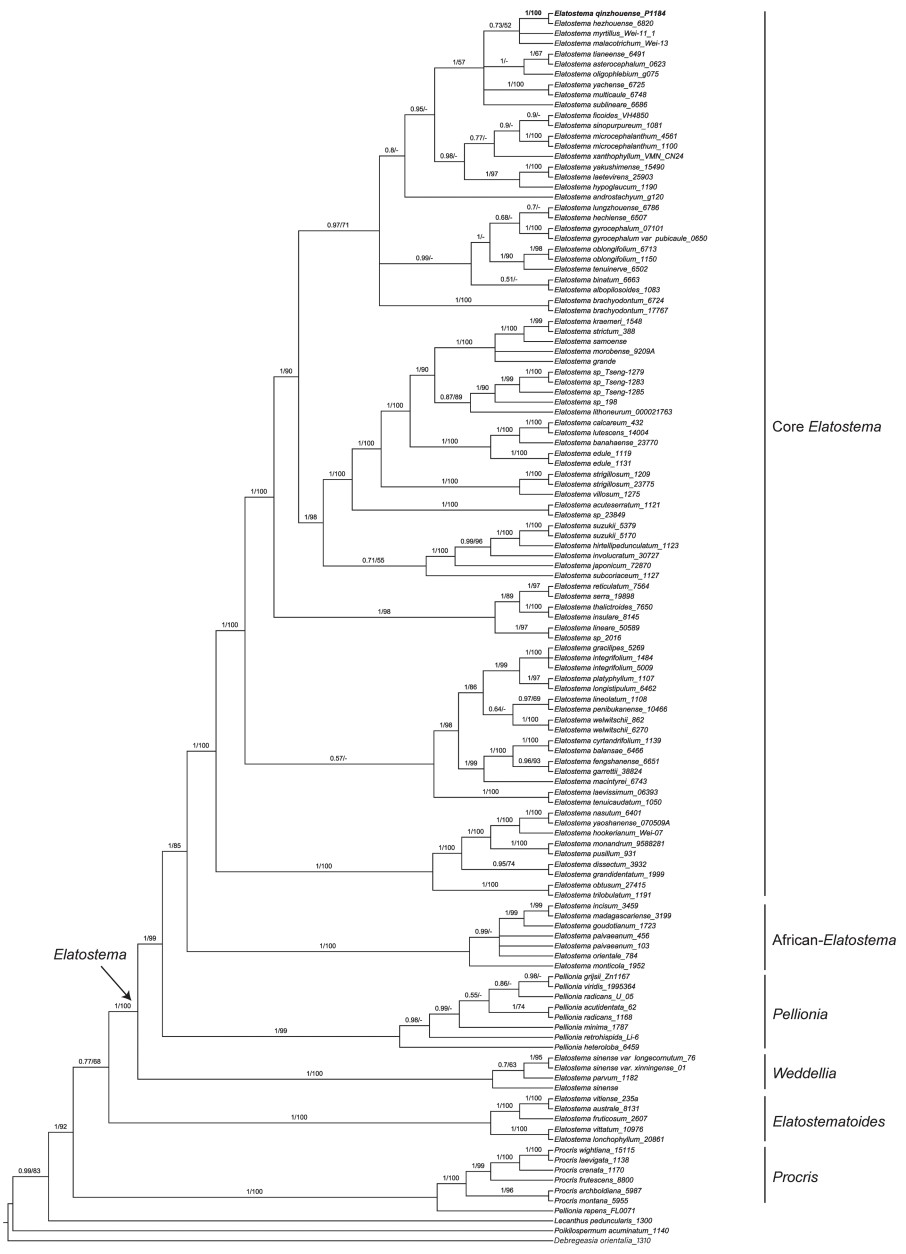

**Figure 2  Phylogenetic tree of *Elatostema* s.l. generated from Bayesian Inference (BI) of combined dataset (ITS, *trnH-psbA* and *psbM-trnD*).** Numbers on the branches indicate the posterior probability (≥0.5) of BI and bootstrap values (≥50%) of the ML analyses.

the subapical appendage 1.0 mm, corniculate, green. Pistillate inflorescences solitary, 5–6 mm, bearing more than 40 flowers in a pedunculate receptacle-like involucre; peduncle ca 1.0 × 0.8 mm, glabrous, bracteate; bracts 0.9 mm, deltate, furfuraceous; receptacle-like involucre 1–1.5 × 2.5–3.0 mm, subquadrate, not lobed, glabrous, subtended by marginal bracts, the bracts unequal, dark green, furfuraceous, major bracts 2, 1.2–1.5 mm, broadly ovate, with a raised dorsal midrib, the apex corniculate, minor bracts 8, 1.0–1.2 mm, ovate,

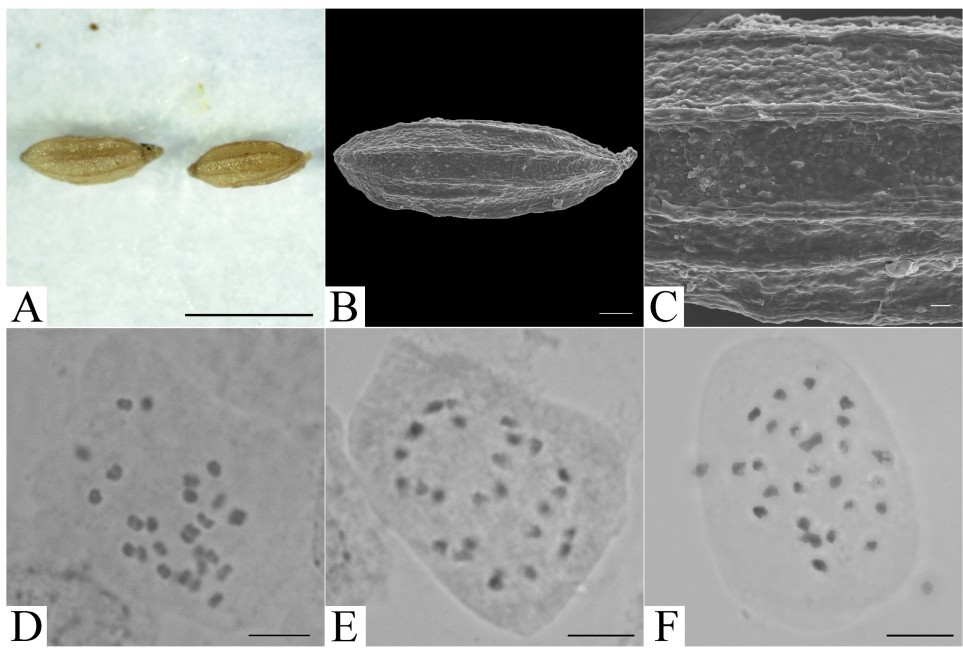

**Figure 3** **Micromorphology of achene and somatic chromosomes at metaphase of *Elatostema qinzhouense*.** (A) Achene from LM observation (Scale bar = 1 mm); (B) achene from SEM observation (Scale bar = 200 μm); (C) achene surface from SEM observation (Scale bar = 20 μm); (D–F) somatic chromosome observation (Scale bar = 10 μm).

**Table 4** **Diagnostic comparison of *Elatostema qinzhouense* and *E. hezhouense*.**

| Characters | E. qinzhouense | E. hezhouense |
|---|---|---|
| leaf laminae | 10–45 × 6–15 mm | 55–115 × 20–25 mm |
| staminate peduncle bract | 1, 1 mm | 2, 3.5 mm |
| pistillate peduncle bract | 0.900 mm | 0.375 mm |
| achene | 0.86–0.94 × 0.27–0.30 | 0.6 × 0.25 |

the apex corniculate. Pistillate flowers pedicellate; pedicel subsessile; bracteoles 2, equal, 1.0–1.3 mm, narrowly lanceolate or linear. Infructescences as pistillate inflorescences; achene 0.86–0.94 × 0.27–0.30 mm, length:width ratio 3.07–0.31:1, oblongoid, pale brown, with 8 narrow longitudinal ridges.

**Distribution and habitat.** *Elatostema qinzhouense* is known from a single locality in Lingshan County, Qinzhou City, Guangxi, China. *E. qinzhouense* is likely calcicolous and grows under evergreen broad-leaved forest on limestone hills. Flowering from December to March, fruiting from March to April.

**Etymology.** *Elatostema qinzhouense* is named after the type locality, Qinzhou City, Guangxi Zhuang Autonomous Region, China.

**Vernacular name.**钦州楼梯草 (Chinese name).

**Additional specimen examined (paratype).** China. Guangxi: cultivated material in Guilin Botanical Garden harvested on 27 January 2019, wild-collected, from Taiping Town,

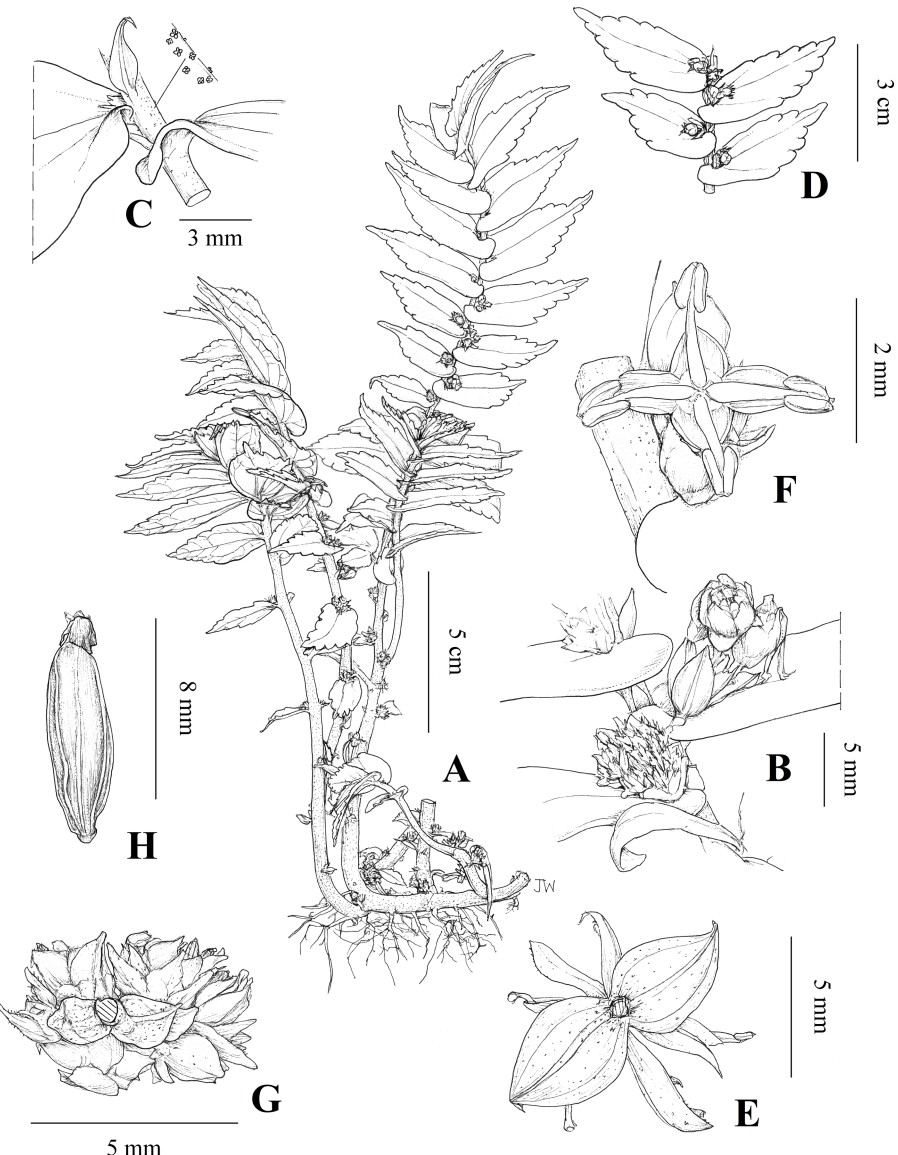

**Figure 4** **Illustration of *Elatostema qinzhouense*.** (A) Habit; (B) staminate and pistillate inflorescences at subsequent nodes of monoecious stem; (C) stipule and furfuraceous stem; (D) leaves; (E) staminate inflorescence viewed from below showing involucre bracts; (F) staminate flower; (G) pistillate inflorescence viewed from below showing involucre bracts; (H) achene. Illustration by Juliet Beentje.

Lingshan County, Qinzhou City, 22.408N, 108.835E (WGS84), elev. 124 m, 26 May 2014, *Pan B and Ma HS P1184* (paratypes BJFC!, CSH!, PE!).

**Conservation Assessment.** *Elatostema qinzhouense* is known from a single locality (AOO 4 km$^2$, criteria B2). At this locality the population of this species comprises ca 200 mature individuals (criteria D1). The only observed population is at the edge of agricultural land on a small limestone hill, which, although deforested in the past, appears not to be actively disturbed. According to *IUCN (2001)* and *IUCN (2019) E. qinzhouense* could

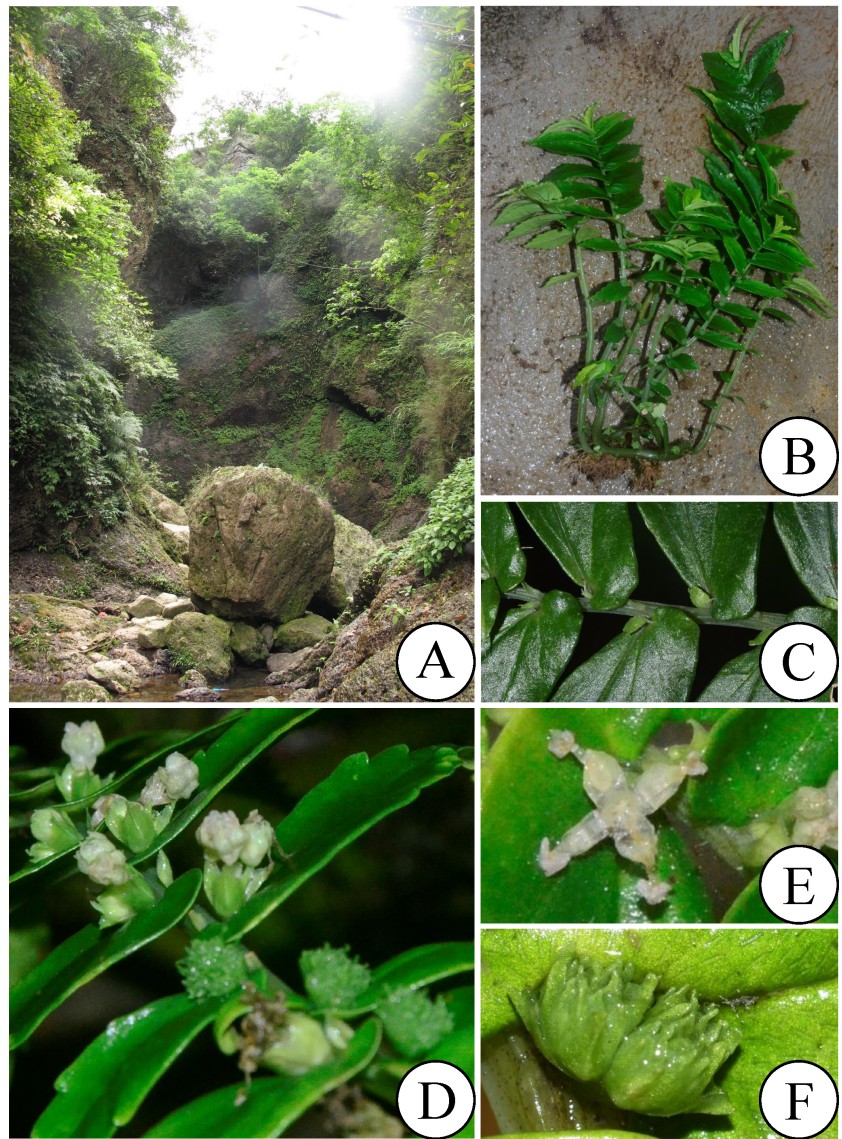

**Figure 5** **Plate of *Elatostema qinzhouense*.** (A) Habitat; (B) habit; (C) leave base and stipule; (D) staminate inflorescence; (E) staminate flower; (F) pistillate inflorescence. Photos by Long-Fei Fu & Bo Pan.

be considered Critically Endangered (CR) according to criteria D1 and B2. Botanically, this part of Guangxi is relatively poorly explored and so it may be that other populations exist that have yet to be observed. Given the frequency of point-endemics in the genus the opposite is also possible. Given the lack of imminent threat and this species resilience to past deforestation, and the frequency of point-endemics amongst karst-associated *Elatostema* suggesting that this species may not have been subject to a large reduction in population size we re-appraise *E. qinzhouense* as Endangered (EN).

## DISCUSSION

All known *Elatostema* species from China have the basic chromosome number $x = 13$ (*Yamashiro et al., 2000*; *Tseng & Hu, 2014*; *Fu et al., 2017b*). Our chromosome count for *E. qinzhouense* of $2n = 26$. *Fu et al. (2017b)* proposed a consistent relationship between ploidy level and reproductive system in *Elatostema* whereby diploid and triploid species are sexual and apomictic respectively. Our observations, therefore, suggest that *E. qinzhouense* is a diploid and reproduces sexually (*Fu et al., 2017b*).

Our phylogenetic analysis of DNA sequence data suggest that *Elatostema qinzhouense* belongs to the Core *Elatostema* clade (*Tseng et al., 2019*) and that it is most closely related to *E. hezhouense*, a cave dwelling species endemic to Guangxi (*Wei, Monro & Wang, 2011*). Geographically, the localities of two species are more than 400 km apart which suggest that there is little opportunity for gene flow as they are wind-pollinated and inhabit deeply dissected mountainous terrain. Morphologically, *E. qinzhouense* is most similar to *E. hezhouense* from which it differs by having smaller leaf laminae, fewer and smaller staminate peduncle bracts, longer pistillate peduncle bracts and a larger achene (see Table 4).

It is worth noting that one of the diagnostic characters for *E. qinzhouense*, the morphology of the peduncle bract, has not been included in previous revisions of Chinese *Elatostema* (*Wang, 1980*; *Wang & Chen, 1995*; *Lin, Friis & Wilmot-Dear, 2003*; *Wang, 2014*). This character has been useful for the characterization of species of *Pilea* (*Monro, 2001*) and on this basis was applied to *Elatostema* (*Wei, Monro & Wang, 2011*). *Tseng et al. (2019)* demonstrated that characters used by *Wang (1980)* and *Wang (2014)* to delimit infrageneric classes in the genus, whilst useful for species identification, were not phylogenetically informative and so exploring additional, potentially informative morphological characters, such as peduncular bract, for species delimitation and infrageneric classification should be a priority.

## CONCLUSIONS

This study is the first attempt to confirm and describe a new species of *Elatostema* based on a combination of morphological, molecular and cytological evidence. The reported plastid genome and chromosome number provide informative data to support further studies on the systematics, evolution and conservation of the genus. We propose that recognizing and describing new species based on the integration of morphological, molecular and cytological data; observations will result in more robust and rational taxa (*Hong, 2016*).

## ACKNOWLEDGEMENTS

We are grateful to Hu-Sheng Ma (GXIB) for joining the fieldtrip, Yu-Jing Wei (GXIB) for SEM experiments and Juliet Beentje (K) for the illustration.

### Funding

This work was supported by the National Natural Science Foundation of China (31860042), the Light of West China Program of the Chinese Academic of Sciences ([2020]59), the Guangxi Natural Science Foundation Program (2017GXNSFBA198014) and the Botanical Illustration fund of RBG Kew. The funders had no role in study design, data collection and analysis, decision to publish, or preparation of the manuscript.

### Grant Disclosures

The following grant information was disclosed by the authors:
National Natural Science Foundation of China: 31860042.
Light of West China Program of the Chinese Academic of Sciences: [2020]59.
Guangxi Natural Science Foundation Program: 2017GXNSFBA198014.
Botanical Illustration fund of RBG Kew.

### Competing Interests

The authors declare there are no competing interests.

### Author Contributions

- Longfei Fu conceived and designed the experiments, performed the experiments, analyzed the data, prepared figures and/or tables, authored or reviewed drafts of the paper, and approved the final draft.
- Alexandre K. Monro conceived and designed the experiments, analyzed the data, prepared figures and/or tables, authored or reviewed drafts of the paper, and approved the final draft.
- Tiange Yang performed the experiments, analyzed the data, prepared figures and/or tables, and approved the final draft.
- Fang Wen and Zibing Xin performed the experiments, prepared figures and/or tables, and approved the final draft.
- Bo Pan analyzed the data, prepared figures and/or tables, conducted field trips, and approved the final draft.
- Zhixiang Zhang and Yigang Wei conceived and designed the experiments, authored or reviewed drafts of the paper, and approved the final draft.

### Field Study Permissions

The following information was supplied relating to field study approvals (i.e., approving body and any reference numbers):

All the collecting locations of the new species reported in this study are outside any natural conservation area and no specific permissions were required for these locations. Since the species are currently undescribed, they are not currently included in the China Species Red List (Wang & Xie, 2004). Our field studies did not involve any endangered or protected species. No specific permits were required for the present study.

## DNA Deposition

The following information was supplied regarding the deposition of DNA sequences:

The complete Elatostema qinzhouense plastid genome sequence is available at GenBank: MW172521.

The Elatostema qinzhouense ribosomal DNA sequence is also available at GenBank: MW172522.

## Data Availability

Raw Elatostema qinzhouense reads are available at NCBI SRA: SRR13189611.

The voucher specimen number is B Pan & HS Ma P1184, collected from China and deposited at IBK.

## New Species Registration

The following information was supplied regarding the registration of a newly described species:

*Elatostema qinzhouense*: 77215696-1.

## Supplemental Information

Supplemental information for this article can be found online at http://dx.doi.org/10.7717/ peerj.11148#supplemental-information.

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
