# Peer review of "Elatostema qinzhouense* (Urticaceae), a new species from limestone karst in Guangxi, China"

_PeerJ, doi:10.7717/peerj.11148_

## Round 0.1 · original submission · Minor Revisions

The authors of the work should take into account the comments of the reviewers to prepare the manuscript before being accepted for publication.

·

Basic reporting

1. Basic reporting

The article is generally well written and the English is excellent. All together it is a very professionally prepared article. The text is self-contained with regard to presentation of background, observations, discussion and conclusions. The literature treatment is admirable. The reviewer will only make a few suggestions that may improve the text.

Abstract: The results are presented in a sequence that starts with information about the plastid genome. That is somehow surprising for the description of a new species that must initially have been identified – or suggested to be a new species - on the basis of morphological observations. In the abstract, the morphological information comes as the very last words in the abstract. The abstract would agree better with the main text of the paper if it mentioned morphological characters that initially supported the idea that the species was indeed a new species, and that this is followed by the information about the observations of the DNA. And then the abstract may conclude with information about the cytology and the phylogenetic analysis.

Introduction: In line 50 it is stated that Elatostema is one of two species-rich genera in the Urticaceae with succulent species … It would be useful if it was stated that the other must be Pilea.

In line 65-69 it is mentioned that there is a risk of over-description of new species in Elatostema. That is certainly true, and it is certainly also true that phylogenies based on DNA sequence data may help to identify closely related species. But it is not stated to what extent these phylogenies are now adequate to so to speak “weed out” over-described new species. See further under comments on the diagnosis in line 197-200 and the discussion in line 258-272.

The data deposit statements seem adequate, and the reviewer could address the plastid genome data.

In line 75-79 it is briefly described how the new species was discovered, but there is only a very short description of how it was realized that the population of sterile plants – once they flowered in the Guilin Botanical Garden – were identified as a new species and the detailed DNA sequence data was initiated. The text about the field study permits seems appropriate.

Materials & Methods: An important point about nomenclature, particularly new names published in entirely digital media, is to make sure that the digital repositories are permanent. Perhaps the relevant articles of the International Code of Nomenclature for algae, fungi and plants should be cited and the code itself referred to (https://www.iapt-taxon.org/nomen/main.php - Turland, N. J., Wiersema, J. H., Barrie, F. R., Greuter, W., Hawksworth, D. L., Herendeen, P. S., Knapp, S., Kusber, W.-H., Li, D.-Z., Marhold, K., May, T. W., McNeill, J., Monro, A. M., Prado, J., Price, M. J. & Smith, G. F. (eds.) 2018: International Code of Nomenclature for algae, fungi, and plants (Shenzhen Code) adopted by the Nineteenth International Botanical Congress Shenzhen, China, July 2017. Regnum Vegetabile 159. Glashütten: Koeltz Botanical Books. DOI https://doi.org/10.12705/Code.2018). It is also relevant to point out whether the requests of recommendations under Article 29 are met with (Rec. 29A.2. Authors of electronic material should give preference to publications that are archived and curated, satisfying the following criteria as far as is practical (see also Rec. 29A.1): (a) The material should be placed in multiple trusted online digital repositories, e.g. an ISO-certified repository. (b) Digital repositories should be in more than one area of the world and preferably on different continents.)

Apart from this, the nomenclature seems to follow PeerJ s requirements for its new species policies, and – when the Diagnosis and Description are improved as suggested here – they will meet the requirements of the International Code of Nomenclature for algae, fungi, and plants (Shenzhen Code).
Results / Diagnosis: It does indeed seem as the species described here is indeed new, although closely related to E. hezhouense. Line 197-200, and Discussion: Line 252-272. The morphological characters that are said to be distinctive when compared with E. hezhouense do seem small. It is stated under Materials & Methods, line 101-105 that the same criterial as with other taxonomic treatments are used. That seems all right, although the morphological characters separating the new species from E. hezhouense are apparently slight, but a short mentioning of phytogeographical facts, for example under Discussion: Line 252-272, would seem a usefull supplement. It does not seem that the subspecies concept is much used for very local Chinese taxa of Elatostema (whereas varieties are sometimes used). Could it not be worth to compare the geographical position and habitat in the localities where the new species and E. hezhouense are found? AS far as the reviewer could find out, the type locality of the new species and the type locality of E. hezhouense are more than 400 km apart. Of course, the new species can always be sunk to the level of subspecies later, should further studies suggest that this would be a suitable taxonomic conclusion.

Further about the Diagnosis (line 197-198): The wording of the diagnosis should be improved from “Most similar to Elatostema hezhouense from which it differs by the smaller leaf laminae (28-42 x 10-15 mm vs. 55-115 x 20-25 mm), …” to “Most similar to Elatostema hezhouense from which it differs by the smaller size of the major leaf laminae (28-45 x 10-15 mm vs. 55-115 x 20-25 mm), …” The current wording was not clear to the reviewer; it could superficially look as if the sizes referred to the size of the smaller laminae (in the description called “minor laminae”).

It added to the confusion that the size of the “minor laminae” are not described in the protologue of Elatostema hezhouense; are “minor laminae” not developed in Elatostema hezhouense? If so, that would add another diagnostic character? The “minor laminae” are easily seen in Fig. 4B and 4C of the new species, but “minor laminae” cannot be seen in the drawings of Elatostema hezhouense accompanying the protologue. – In addition, the width of the “major laminae” in the new species is indicated as 28-42 mm in the Diagnosis and in Table 4, but 28-45 mm in the Description.

In the Description (line 203) the text about stipules is not clear. It says: “Stipules solitary, interpetiolar, opposite the leaf at each node, …” But the “major laminae” and the “minor laminae” are not strictly opposite, so the stipules cannot be interpetiolar (meaning located between the petioles); the correct word must be intrapetiolar, which means placed in the axil between the leaf and the stem, and – if this is correct – the wording would be more clear if it is changed to: “Stipules solitary, intrapetiolar, in the leaf-axil at each node, …” The reviewer could not spot the stipules in Fig. 4.

Distribution and habitat (line 228-231): As mentioned above, it would improve the paper if the geographical positions and habitats of the new species and Elatostema hezhouense were compared.

The figures are all relevant and of adequate quality.

Experimental design

2. Experimental design

Strictly speaking, there is no experimental design in the paper, but the laboratory work seems to have be carried out with rigorous and careful investigations, and the methods are adequately described. The paper represents original primary research within the scope of PeerJ.

The research questions are well defined (but see the comments on several parts of the text, where it could seem as the morphological observations were used to test the DNA data, while the sequence of event would seem to have been the opposite).
A few additional comparisons of the localities and habitats of the new species and Elatostema hezhouense are proposed.

The methods are described in sufficient detail to allow replication of the study.

Validity of the findings

3. Validity of the findings
The findings all appear to be valid, but could in a few cases be supplemented with additional discussion (see Basic reporting).
All necessary data are well provided, but in a few places rewording of the presentation would be an improvement.
The conclusions are well stated.

Additional comments

No further comments. All is includes in 1. Basic reporting.

Reviewer 2 ·

Basic reporting

The manuscript is well prepared in the text with sufficient literature and reasonable structure too. Some words should be modified so that the readers can get the meanings very clearly. Please see the highlighted words.

Experimental design

The methods are well designed and the original data is comprehensively analysed in the whole manuscript too.

Validity of the findings

The new finding of the new species can increase our knowledge about the biodiversity of the genus Elatostema. Authors apply integrated methods to elucidate the new taxa and its phylogenetic relationship, which provided a well understanding about the family of Urticaceae. However, the description about the morphological similarity and the close relationship should be careful because these are probably NOT so. Please see the annoted PDF file.

Additional comments

The manuscript about the new species of Elatostema qinzhouense was well prepared and the result is convincing too, on basis of the morphological and whole plastid genome evidences. But several minor doubts should be cleaned before publication (Pls see the reviewed PDF file).

Annotated reviews are not available for download in order to protect the identity of reviewers who chose to remain anonymous.

Reviewer 3 ·

Basic reporting

The paper by Longfei Fu et al. "Elatostema qinzhouense (Urticaceae), a new species from
limestone karst in Guangxi, China" contains the description of a new species in the genus Elatostema. Authors have done a thorough work besides the usual morphological description by including cytological and molecular data as well, all suggesting a well defined, different taxon. While the plant material on which the new taxon was described was collected outside protected areas and therefore collecting permits were not necessary, the authors have fullfilled with the standard requirements in depositing vouchers in known herbaria and copies of the work in digital repositories.
I would like to ask the authors to add as an additional figure a good resolution image of the holotype (IBK00426150). Also, it would be useful to know the number of specimens on which the measurements depicted in Table 4 are based, to evaluate the robustness of the differences.

Experimental design

Morphological, molecular and cytological methods are correct. As expressed before, I would like to know the number of specimens on which the measurements have been done.

Validity of the findings

No comment

Additional comments

This is a very well done work and besides the two minimal observations I would suggest to accept it for publishing in PeerJ. My congratulations to the authors.

---

## Round 0.2 · accepted · Accept

The work is ready for publication in its current form, the authors have made all the necessary improvements according to the recommendations of the reviewers.